

# Retrieval of aerosol microphysical properties from atmospheric lidar sounding: an investigation using synthetic measurements and data from the ACEPOL campaign

William G. K. McLean[1], Guangliang Fu[1], Sharon P. Burton[2], and Otto P. Hasekamp[1]

[1]Netherlands Institute for Space Research (SRON, NWO-I), Utrecht, the Netherlands
[2]NASA Langley Research Center, Hampton, VA, USA

**Correspondence:** Will McLean (williamgkm@sron.nl)

**Abstract.**

This study presents an investigation of aerosol microphysical retrievals from High Spectral Resolution Lidar (HSRL) measurements. Firstly, retrievals are presented for synthetically-generated lidar measurements, followed by an application of the retrieval algorithm to real lidar measurements. Here, we perform the investigation for an aerosol state vector that is typically

used in multi-angle polarimeter (MAP) retrievals, so that the results can be interpreted in relation to a potential combination of lidar and MAP measurements. These state vectors correspond to a bimodal size distribution, where column number, effective radius, and effective variance of both modes are treated as fit parameters, alongside the complex refractive index and particle shape. The focus is primarily on a lidar configuration based on that of the High Spectral Resolution Lidar-2 (HSRL-2), which participated in the ACEPOL (Aerosol Characterization from Polarimeter and Lidar) campaign, a combined project between

NASA and SRON (Netherlands Institute for Space Research). The measurement campaign took place between October and November 2017, over the western region of the USA. Six different instruments were mounted on the aeroplane: four MAPs, and two lidar instruments: HSRL-2, and the Cloud Physics Lidar (CPL). Most of the flights were carried out over land, passing over scenes with a low aerosol load. One of the flights passed over a prescribed forest fire in Arizona on the 9th of November, with a relatively higher AOD, and it is the data from this flight that is focussed on in this study. A retrieval of the aerosol mi-

crophysical properties of the smoke plume mixture was attempted with the data from HSRL-2, and compared with a retrieval from the MAPs carried out in previous work pertaining to the ACEPOL data.

The synthetic HSRL-2 retrievals resulted for the fine mode in a mean absolute error (MAE) of 0.038 (0.025) μm for the effective radius, 0.052 (0.037) for the real refractive index, 0.010 ($7.20 \times 10^{-3}$) for the imaginary part of the refractive index, 0.109 (0.071) for the spherical fraction, and 0.054 (0.039) for the AOD at 532 nm, where the retrievals inside brackets indicate

the MAE for noise-free retrievals. For the coarse mode, we find the MAE is 0.459 (0.254) μm for the effective radius, 0.085 (0.075) for the real refractive index, $2.06 \times 10^{-4}$ ($1.90 \times 10^{-4}$) for the imaginary component, 0.120 (0.090) for the spherical fraction, and 0.051 (0.039) for the AOD. A study of the sensitivity of retrievals to the choice of prior and first guess showed that, on average, the retrieval errors increase when the prior deviates too much from the truth value. These experiments revealed that the measurements primarily contain information on the size and shape of the aerosol, along with the column number. Some





information on the real component of the refractive index is also present, with the measurements providing little on absorption or on the effective variance of the aerosol distribution, as both of these were shown to depend heavily on the choice of prior.

Retrievals using the HSRL-2 smoke-plume data yielded, for the fine mode, an effective radius of 0.107 µm, a real refractive index of 1.561, an imaginary component of refractive index of 0.010, a spherical fraction of 0.719, and an AOD at 532 nm of 0.505. Additionally, the single-scattering albedo (SSA) from the HSRL-2 retrievals was 0.940. Overall, these results are in
good agreement with those from the SPEX and RSP retrievals.

# 1   Introduction

Aerosols play a key role in the climate of the Earth, and cause a direct climate forcing through absorbing and reflecting incoming shortwave radiation. Aerosol particles also indirectly modify the climate via their effect on cloud properties (Seinfeld and Pandis, 2016; Hansen et al., 1998; Haywood and Boucher, 2000). The first indirect effect, referred to as the Twomey effect
(Twomey, 1974), is the effect that a larger quantity of aerosol particles results in an increased number of smaller droplets, increasing the reflectivity of the cloud. Smaller droplets also reduce the precipitation efficiency, thereby increasing the lifetime of clouds (Albrecht, 1989). Both the direct and indirect climate forcings are still not all that well understood, and may be the greatest source of errors in attempting to make future projections about climate change (Andreae et al., 2005).

The presence of aerosols in the atmosphere can have detrimental effects on air quality, and thereby human health. Aerosols
are released into the atmosphere from a multitude of sources, though can be classified broadly into two categories: those originating from natural means, and those resulting from anthropogenic influences on the planet. For example, sea salt and dust are amongst those aerosol particles emitted through natural means, whereas sulphuric acid salts used in industry are illustrative of the aerosols resulting from human influence. Aerosols are transported throughout the atmosphere by means of localised turbulence, and through the direct atmospheric transportation mechanisms. For example, desert dust upwelled from the Sa-
haran region of Africa is often transported across the Atlantic Ocean into the Amazon rainforest (Yu et al., 2015). To further understand the effect on the climate that aerosol particles have, remote sensing measurements from satellite instruments are a necessity, coupled with advanced retrieval methods in order to understand their physical and optical properties. For quantification of the direct effect, measurements of optical properties such as Aerosol Optical Depth (AOD) and Single Scattering Albedo (SSA) are important (Loeb and Su, 2010; Lacagnina et al., 2015, 2017). For quantification of the indirect effect, mea-
surements of aerosol column number and size are needed (Dusek et al., 2006; Hasekamp et al., 2019b) as well as information on the vertical profile, and measurements as close to the cloud as possible (Quaas et al., 2020).

It has been well established that polarimetric remote sensing measurements can be used to achieve high-accuracy retrievals of microphysical properties of an aerosol column. When carrying out passive remote sensing, observations from instruments mounted on satellites providing both intensity and degree of linear polarisation at multiple viewing angles contain the great-
est wealth of information pertaining to aerosols in the atmosphere (Mishchenko and Travis, 1997a; Hasekamp and Landgraf, 2007). This is due to the fact that the angular dependent scattering matrix elements of the aerosol particles are very sensitive to variations in the aerosol microphysical properties, such as the complex refractive index, particle size, and particle distribu-





tion (Hansen and Travis, 1974; Mishchenko and Travis, 1997b). Additionally, the linear polarisation measurement is mainly comprised of radiation scattered one time, thus the most highly polarised signals will mainly be the results of single-scattering.

This means that the scattering matrix characterising the aerosol particles will largely remain preserved. Successful aerosol retrievals and applications from satellite based polarimetric measurements have been performed from POLDER by Dubovik et al. (2011); Hasekamp et al. (2011); Lacagnina et al. (2015, 2017); Chen et al. (2018, 2019); Hasekamp et al. (2019a).

Measurements using high spectral resolution lidar (HSRL) techniques can provide constraints on aerosol optical and/or microphysical properties, placing them into broad categories, such as differentiating between aerosols from naturally arising

processes, and anthropogenically emitted aerosols (Burton et al., 2012). Although the information on aerosol microphysical and optical properties in lidar measurements is more limited than in multi-angle polarimetric measurements, an important asset of lidar measurements is that they provide information as a function of altitude. An example of such an instrument is the Cloud-Aerosol Lidar with Orthogonal Polarization (CALIOP) elastic backscattering lidar, which has been in operation since 2006 (Winker et al., 2010). CALIOP measures the attenuated backscatter coefficient at 532 nm and 1064 nm, along with the

depolarisation ratio at 532 nm ($2\beta + \delta^{\mathrm{pol}}$). The latest generation of lidar instruments utilise the high spectral resolution lidar method (Hair et al., 2008). The HSRL technique can measure aerosol extinction as a function of altitude. The HSRL techniques reduce errors in the measurements, and additionally provides an improved measurement of the aerosol depolarisation ratio (Burton et al., 2012), which is an important parameter for determining the particle shapes in an aerosol mixture. An instrument with such a setup is the ATLID instrument, which is planned for the upcoming ESA Earthcare mission (Illingworth et al.,

2015). ATLID will measure the extinction, backscatter, and the depolarisation ratio at 355 nm ($\alpha+\beta+\delta^{\mathrm{pol}}$). The Cloud-Aerosol Transport System (CATS) instrument operated onboard the ISS from 2015-2017. A problem with the laser however precluded the testing of the HSRL method on this instrument (Yorks et al., 2014; Yorks et al., 2016). The measurement configuration used for most of the CATS observations was attenuated backscatter and depolarisation at 532 nm and 1064 nm ($2\beta + 2\delta^{\mathrm{pol}}$).

A combined measurement vector of three backscatter coefficients and two extinction coefficients ($3\beta + 2\alpha$) is generally taken

as the minimum level of information required in order to achieve sufficiently accurate retrievals of microphysical properties (Müller et al., 1998; Böckmann et al., 2005; Veselovskii et al., 2002; Burton et al., 2016). This result was found from multiple sets of ground-based lidar measurements (Müller et al., 1999; Böckmann, 2001; Donovan and Carswell, 1997). Several investigations of how much information can be extracted from multi-wavelength lidar measurements of aerosol optical properties have been carried out recently. Tesche et al. (2019) present retrievals from a measurement vector containing the backscatter

coefficient at 355, 532, and 1064 nm and the extinction coefficient at 355 and 532 nm, with a measurement of the depolarisation ratio added at one wavelength, to try and determine the most optimal combination of input parameters to a retrieval. Using different combinations of depolarisation ratio in the input, to account for the contribution of the non-spherical particles, Tesche et al. (2019) test the performance of the inversion using the model of Dubovik et al. (2006) for mixed dust scenarios, which is representative of aerosol scenarios such as long-range transport of mineral dust. They concluded that, dependent on

the limitations of the instrument used, the standard input for inversion of lidar data with mineral dust particles present using the spheroid model of Dubovik et al. (2006) should be the $3\alpha + 2\beta$ setup, with the addition of the depolarisation ratio ($\delta^{\mathrm{pol}}$) at as many wavelengths as possible. A multitude of studies were carried out on the retrieval of aerosol microphysical properties





from the inversion of lidar measurements in in the framework of the previous NASA Aerosol-Cloud-Ecosystem (ACE) mission (Pérez-Ramírez et al., 2013; Veselovskii et al., 2013; Chemyakin et al., 2014, 2016; Whiteman et al., 2018; Pérez-Ramírez et al., 2019; Pérez-Ramírez et al., 2020).

Burton et al. (2016) use a forward model look-up table approach on a $3\beta + 2\alpha$ measurement combination to represent spherical particles for the determination of measurement sensitivities to assumptions and constraints in retrievals. They considered a mono-modal aerosol size distribution with the effective radius, effective variance, complex refractive index, and the particle number concentration as unknown parameters. Burton et al. (2016) find that the $3\beta + 2\alpha$ combination of measurements provides about 3-4 independent pieces of information, and only limited information about aerosol absorption.

In this work, we build further on the study of Burton et al. (2016) and extend it in the following aspects: i) We use a retrieval state vector that is based on a bimodal size distribution as well as non-spherical particles, instead of a mono-modal size distribution with spherical particles. Our choice of state vector is in line with the aerosol description typically used in Multi-Angle Polarimeter (MAP) retrievals (Waquet et al., 2009; Hasekamp et al., 2011; Wu et al., 2015, 2016), so that the results can be interpreted in relation to potential combination of lidar and MAP measurements. ii) Instead of performing a linear uncertainty/information content analysis, we apply an iterative retrieval scheme, that can be used to perform actual retrievals. Based on synthetic observations, we first investigate the retrieval capability of two types of lidar, a so-called "super lidar" that operates with measurements of extinction coefficient, backscatter coefficient, and depolarisation ratio at 355, 532, and 1064 nm ($3\alpha$, $3\beta$, and $3\delta^{pol}$), similar to the measurement configuration used by Haarig et al. (2018), and the HSRL-2 instrument, which provides the same except for the extinction at 1064 nm, thus has a $2\alpha$, $3\beta$ and $3\delta^{pol}$ configuration. The dependence on the retrieval of the chosen a priori values of the microphysical parameters was assessed. Finally, we apply our retrieval scheme to real HSRL-2 measurements obtained during the Aerosol Characterization from Polarimeter and Lidar (ACEPOL) campaign, during which HSRL-2 was mounted onboard the NASA Earth Resources-2 (ER-2) high altitude (approximately 20 km) aircraft. The ACEPOL flights were carried out from October to November 2017, beginning from the NASA Armstrong airbase in Palmdale, California, USA. In addition to HSRL-2, and the other lidar instrument onboard (Cloud Physics Lidar (CPL) (McGill et al., 2002)), four polarimeters were also part of the payload: The Spectropolarimeter for Planetary Exploration (SPEX airborne) (Smit et al., 2019), the Research Scanning Polarimeter (RSP) (Cairns et al., 2004), the Airborne Multi-angle SpectroPolarimetric Imager (AirMSPI) (Diner et al., 2013), and the Airborne Hyper-Angular Rainbow Polarimeter (AirHARP) (Martins et al., 2018). HSRL-2 returned height-resolved profiles of the backscatter coefficient and the depolarisation ratio at three wavelength, 355 nm, 532 nm, and 1064 nm, and the extinction coefficient at 355 nm and 532 nm. CPL measured the vertically resolved attenuated backscatter coefficient at 355 nm, 532 nm, and 1064 nm, and the depolarisation ratio at 1064 nm. This study considers the column integrated values of extinction coefficient (i.e. the aerosol optical depth (AOD)), backscatter coefficient, and depolarisation ratio.

The rest of this paper is structured as follows: Section 2 gives an overview of the retrieval algorithm used, and defines the various quantities used in the retrievals. Section 3 outlines the measurements in the study, firstly describing the synthetic data, followed by the HSRL-2 measurements used in the real data retrievals. Also described in this section are the error models used in all retrievals. Section 4 then presents the results of the retrievals, beginning with those from synthetic measurements in Sect.





4.1, followed by retrievals from ACEPOL lidar measurements in Sect. 4.2. Finally, Sect. 5 summarises the results, and presents the conclusions of this study.

## 2 Methodology

### 2.1 Retrieval algorithm

Here, the method used in retrieving the microphysical aerosol parameters from lidar measurements of optical properties is described. The vector $\boldsymbol{y}$ represents the lidar measurements, which contains the aerosol layer optical depth, layer-integrated depolarisation ratio, and the layer-integrated backscatter coefficient (or a selection) at different wavelengths. These are all defined in Sect. 2.2. The microphysical parameters to be retrieved are contained in the state vector, $\boldsymbol{x}$. The retrieval of $\boldsymbol{x}$ from the measurement vector $\boldsymbol{y}$ requires a forward model, $\mathbf{F}$. Thus we have the relationship:

$$\boldsymbol{y} = \boldsymbol{F}(\boldsymbol{x}) + \boldsymbol{e}_y, \tag{1}$$

where the error on the measurements is represented by the term $\boldsymbol{e}_y$. In the two-mode aerosol retrieval used in this study, the fine and coarse aerosol modes are denoted by the subscripts "f", and "c" respectively. Each of the aerosol modes is described by a log-normal function, and are characterised by the effective radius $r_{\mathrm{eff}}^{\mathrm{f;c}}$, the effective variance $v_{\mathrm{eff}}^{\mathrm{f;c}}$, the real and the imaginary parts of the refractive index $m_{\mathrm{r}}^{\mathrm{f;c}}$, and $m_{\mathrm{i}}^{\mathrm{f;c}}$, the aerosol column number $N^{\mathrm{f;c}}$, and the fraction of spheres $f_{\mathrm{sphere}}^{\mathrm{f;c}}$. The refractive indices themselves are not directly included in the state vector. As per recent work (Fu et al., 2020; Fu and Hasekamp, 2018), the spectrally-dependent refractive indices were retrieved via a parameterisation, constructing the complex refractive index at each wavelength via

$$m^{\mathrm{f;c}}(\lambda) = \sum_{k=1}^{n_\alpha^{\mathrm{f;c}}} \alpha_k^{\mathrm{f;c}} m^{k,\mathrm{f;c}}, \tag{2}$$

where the mode component coefficients $\alpha_k^{\mathrm{f;c}}$ are the parameters included in the state vector. The fixed spectrally-dependent complex refractive indices for each component are prescribed as in the standard types of D'Almeida et al. (1991) (inorganic/sulphate, black carbon, and dust), shown in Table 1. In this work, we set $n_\alpha^{f;c} = 2$, and make the assumption that the fine and the coarse mode respectively consist of inorganic and black carbon components, and dust and inorganic components. An advantage to this parameterisation of the refractive indices is that it accounts for spectral dependence, and also eliminates the broad order-of-magnitude range that the imaginary component can occupy. With this parameterisation, the total number of parameters free to vary in the retrieval is six for each mode, thereby twelve in total for the bimodal setup we use.

The retrieval of the state vector from the measurements was achieved with the application of a damped Gauss-Newton iteration method alongside Phillips-Tikhonov regularisation (Fu and Hasekamp, 2018). The inversion algorithm finds the solution $\hat{\boldsymbol{x}}$, solving the minimisation-optimisation problem:

$$\boldsymbol{x} = \min_{\boldsymbol{x}} \left( [\boldsymbol{F}(\boldsymbol{x}) - \boldsymbol{y}]^T \, \boldsymbol{S}_y^{-1} \, [\boldsymbol{F}(\boldsymbol{x}) - \boldsymbol{y}] \right) + \left( [\boldsymbol{x} - \boldsymbol{x}_a]^T \, \gamma^2 \boldsymbol{W}^{-1} \, [\boldsymbol{x} - \boldsymbol{x}_a] \right), \tag{3}$$





**Table 1.** Refractive indices as a function of wavelength.

|  | 355 nm | | 532 nm | | 1064 nm | |
|---|---|---|---|---|---|---|
|  | $m_r$ | $m_i$ | $m_r$ | $m_i$ | $m_r$ | $m_i$ |
| Black carbon | 1.75 | 0.70 | 1.75 | 0.70 | 1.76 | 0.70 |
| Inorganic | 1.50 | $1.00\times10^{-7}$ | 1.50 | $1.00\times10^{-7}$ | 1.50 | $1.00\times10^{-7}$ |
| Dust | 1.53 | $1.66\times10^{-2}$ | 1.53 | $6.33\times10^{-3}$ | 1.53 | $1.08\times10^{-3}$ |

Here, $S_y$ is the measurement error covariance matrix, which is discussed further in Sect. 3. The diagonal matrix $W$ contains weighting factors for the elements of the state vector in the side constraint provided by the a priori state vector, represented by $x_a$, ensuring that all of the state vector parameters remain the same order of magnitude. Assigning a weight to each state vector

component also enables one to allow more flexibility in the retrieval to certain parameters (Hasekamp et al., 2011). Table 2 gives the weights and priors used in all of the retrievals carried out in this study. The weights were chosen based on values used in Fu et al. (2020), with some minor adjustments.

The forward model computes the aerosol optical properties from the microphysical parameters. The Mie T-matrix geometrical optics database of Dubovik et al. (2006) is used, along with the proposed distribution of spheroid aspect ratios in order

to calculate the optical properties for an aerosol distribution containing a mixture of spheroids and spherical particles, with the fraction of spherical particles a free parameter. In addition to the underdeterminedness of the inversion problem, limitations to the forward model can further inhibit the retrieval of the microphysical properties. Veselovskii et al. (2016) found that the spheroid model struggles to replicate values of depolarisation ratio commensurate to those of pure dust. Several other studies have remarked on the limitations of the dust model, especially when retrieving properties for particles with a real refractive

index larger than 1.5 (Müller et al., 2012; Shin et al., 2018; Tesche et al., 2019; Kahnert et al., 2020)

As the nature of the forward model is nonlinear, the inversion problem is solved in an iterative manner, through replacing the forward model for iteration step $n$ with the Taylor expansion up to first order:

$$F(x) \approx F(x_n) + K(x - x_n). \tag{4}$$

Here, $K$ is the Jacobian matrix (where $K_{ij} = \frac{\partial F_i}{\partial x_j} x_n$), containing the derivatives of the forward model with respect to each

variable of the state vector. The optimisation problem for each iteration step can be written as

$$\tilde{x}_{n+1} = \min_{\tilde{x}} \left( [\tilde{K}\,\tilde{x} - \tilde{y}]^T [\tilde{K}\,\tilde{x} - \tilde{y}] \right) + \gamma^2 \left( [\tilde{x} - \tilde{x}_a]^T [\tilde{x} - \tilde{x}_a] \right), \tag{5}$$

where $\tilde{K} = S_y^{-\frac{1}{2}} K W^{\frac{1}{2}}$, $\tilde{x} = W^{-\frac{1}{2}} x$, and $\tilde{y} = S_y^{-\frac{1}{2}}(y - F(x_n))$.

The solution is given by

$$\tilde{x}_{n+1} = \tilde{x}_n + \Lambda \left[ (\tilde{K}^T \tilde{K} + \gamma^2 I)^{-1} (K^T \tilde{y} - \gamma^2 (\tilde{x}_n - \tilde{x}_a)) \right]. \tag{6}$$



The parameter $\gamma^2$ is a regularisation parameter for the side constraint, which is varied for each iteration between 0.1 and 5, and $\Lambda$ is a filter factor limiting the step size of the state vector for each iteration. Per iteration, $\Lambda$ is varied between 0 to 1 with a step size of 0.1, and for each $\Lambda$ a value of $\gamma^2$ is trialled. Inside this double loop, the forward model is calculated for each combination of $\Lambda$ and $\gamma^2$. The forward model which yields the lowest value of $\chi^2$ is chosen as the best for that iteration. The following iteration then uses the corresponding state vector retrieved from the previous one, to compute a new forward model,

and the process is repeated. At the end of the retrieval, the state vector and forward model with the lowest value of $\chi^2$ is then selected as the result. The goodness of fit check serves to eliminate cases where the first guess state vector deviates too far from the optimum solution, and does not approach it through the retrieval.

The goodness of fit, $\chi^2$, is calculated via:

$$\chi^2 = \frac{1}{n_{\mathrm{meas}}} \sum_{i=1}^{n_{\mathrm{meas}}} \frac{(F_i - y_i)^2}{S_y(i,i)} \tag{7}$$

where $n_{\mathrm{meas}}$ is the total number of lidar measurements (multispectral values of AOD, layer-integrated backscatter coefficient, and layer-integrated depolarisation ratio for the pixel). A criterion of $\chi^2 \leq 5$ was imposed as the filter to determine the success of a retrieval.

In order to quantify the quality of the retrievals that passed the goodness of fit test, the mean absolute error (MAE) and the bias were used. The MAE is given by:

$$\mathrm{MAE} = \frac{1}{n_{\mathrm{pass}}} \sum_{i=1}^{n_{\mathrm{pass}}} |x_i^{\mathrm{retr}}[j] - x_i^{\mathrm{truth}}[j]|, \tag{8}$$

with the bias calculated via

$$\mathrm{bias} = \frac{1}{n_{\mathrm{pass}}} \sum_{i=1}^{n_{\mathrm{pass}}} (x_i^{\mathrm{retr}}[j] - x_i^{\mathrm{truth}}[j]), \tag{9}$$

where the sum is over $n_{\mathrm{pass}}$, the number of pixels that pass the goodness of fit test. The variable $x_i^{\mathrm{retr}}[j]$ represents the $j^{th}$ element of the retrieved state vector for pixel $i$, with $x_i^{\mathrm{truth}}[j]$ denoting the $j^{th}$ element of the true state vector for pixel $i$.

## 2.2 Aerosol optical properties

The lidar measurements of optical properties are dependent on the scattering matrix. Under the general assumptions of: (i) scattering takes place in an assembly of randomly orientated particles, each having a plane of symmetry; (ii) scattering occurs in an assemblage of particles and their mirror particles in equal numbers, and with random orientations; (iii) Rayleigh scattering occurring with or without depolarisation effects (van de Hulst, 1957), the aerosol scattering matrix has a simplified block





**Table 2.** Prior, first guess, and weight for each parameter in the synthetic retrievals.

| Parameter | Prior & first guess | Weight |
|---|---|---|
| Aerosol loading of each mode | 0.0001 | 2.0 |
| Spherical fraction of each mode | 0.5 | 0.25 |
| Fine-mode effective radius | 0.2 | 0.1 |
| Fine-mode effective variance | 0.2 | 0.05 |
| Fine-mode black carbon coefficient | 0.025 | 0.025 |
| Fine-mode inorganic coefficient | 0.95 | 0.1 |
| Coarse-mode effective radius | 1.5 | 1 |
| Coarse-mode effective variance | 0.5 | 0.1 |
| Coarse-mode dust coefficient | 0.5 | 0.1 |
| Coarse-mode inorganic coefficient | 0.5 | 0.1 |

diagonal structure:

$$
F(\Theta) = \begin{pmatrix}
F_{11}(\Theta) & F_{12}(\Theta) & 0 & 0 \\
F_{12}(\Theta) & F_{22}(\Theta) & 0 & 0 \\
0 & 0 & F_{33}(\Theta) & F_{34}(\Theta) \\
0 & 0 & -F_{34}(\Theta) & F_{44}(\Theta)
\end{pmatrix}
\tag{10}
$$

where $\Theta$ is the scattering angle and $F_{11}$ is the phase function.

The linear depolarisation ratio of an aerosol mixture is given by

$$
\delta_{\text{col}}^{\text{pol}} = \frac{F_{11}(180°) - F_{22}(180°)}{F_{11}(180°) + F_{22}(180°)}
\tag{11}
$$

e.g. (Mishchenko et al., 2016). The aerosol extinction coefficient, $\alpha$, is defined as the product of the extinction cross section, $\sigma$, and the aerosol concentration, $N$. Integrating this over a vertical column gives the AOD ($\tau$).

The backscatter coefficient was calculated via

$$
\beta_{\text{bs, int}} = \frac{1}{4\pi} \int\limits_{z_1}^{z_2} \mathrm{d}z \, \alpha \omega F_{11}(180°),
\tag{12}
$$

where the integral is over altitude $z$. The retrievals presented in this study use a combination of column integrated extinction
coefficient (i.e. AOD), depolarisation ratio, and backscatter coefficient, but the results would also apply to properties integrated over a particular vertical layer of the atmosphere.





## 3 Measurements

For this study, we use synthetically-generated optical parameters as measurements for the retrievals considered in the first part
of the paper. The focus of this work is on the HSRL-2 setup, with the synthetic retrievals using a configuration analogous to that

of the HSRL-2 instrument of the ACEPOL campaign, described in more detail in Sect. 3.2, along with the instrumental errors.
Additionally, retrievals are presented for a so-called super-lidar setup, used as a benchmark test with the greatest amount of
information available from high spectral resolution lidar measurements. Also shown are results from retrievals with measure-
ment configurations representing simplified versions of CPL and ATLID. In the second part of the paper, lidar retrievals from
real HSRL-2 data taken during the ACEPOL campaign are presented, and compared with measurements from SPEX airborne

and the Research Scanning Polarimeter (RSP).

### 3.1 Synthetic data

In order to investigate the potential of lidar measurements in yielding aerosol microphysical properties, a suite of experiments
carried out on synthetic data were performed. The measurement configuration for the super-lidar setup is $3\alpha + 3\beta + 3\delta^{\mathrm{pol}}$,
with measurements at all three wavelengths: 355, 532, and 1064 nm. HSRL-2 has a $2\alpha + 3\beta + 3\delta^{\mathrm{pol}}$ setup, the same as for

the super-lidar but with the exception of the extinction coefficient at 1064 nm. The CPL-type measurement setup uses the
backscatter coefficient at 355, 532, and 1064 nm, and the depolarization ratio at 1064 $(3\beta + \delta^{\mathrm{pol}})$. The real CPL instrument
returns the attenuated backscatter coefficient, whilst the backscatter coefficient used throughout this work, including both the
forward model and the simulated data, is uncorrected for atmospheric attenuation. Thereby, the forward model used is not
strictly correct for what is returned for the real CPL instrument, but is still sufficient to give a rough estimate of the information

content of such a measurement in terms of the aerosol microphysics. Finally, ATLID returns extinction, backscatter, and
depolarisation ratio at 355 nm $(\alpha + \beta + \delta^{\mathrm{pol}})$.

Synthetic measurements have been created for a two mode setup: a fine mode (denoted by the superscript f) and a coarse
mode (denoted by the superscript c). The synthetic measurements were generated as follows: $r_{\mathrm{eff}}$ was randomly placed in the
range (0.1, 0.3) for the fine mode, and (0.65, 3.4) for the coarse mode. The corresponding ranges of $v_{\mathrm{eff}}$ were (0.1, 0.3), and

(0.4, 0.6), for the fine and coarse mode respectively. The fraction of spheres was placed in the range (0, 1) for both modes.
The column number of each mode was calculated internally from the input parameters, with the input AOD created in the
range (0, 0.7) for the fine mode, and (0, 0.3) for the coarse mode. For the fine-mode refractive indices, values corresponding
to those of black carbon and inorganic aerosols were chosen. The coarse-mode refractive indices were taken to be a mixture
of inorganic and dust particles. The coefficients were chosen such that the real component of the refractive index was in the

range 1.30-1.69. Synthetic retrievals from an HSRL-2 type instrument are the focal point of this study, as this instrumental
configuration represents the most advanced lidar setup currently in use.

We assume the following error values: $\Delta\delta^{\mathrm{pol}}_{355} = 0.001$, $\Delta\delta^{\mathrm{pol}}_{532} = 0.007$, $\Delta\delta^{\mathrm{pol}}_{1064} = 0.007$, with $\Delta\beta$ and $\Delta$AOD both 5% for
all three wavelengths. The error for the depolarisation ratio is an absolute value, and the backscatter coefficient error is a relative
one, multiplied by the measurement. The error on depolarisation ratio was taken from Burton et al. (2015), and the backscatter



error from Burton et al. (2016). We put errors on the synthetic measurements assuming Gaussian noise with standard deviation given by the values above.

## 3.2 ACEPOL HSRL-2 data

The NASA Langley HSRL-2 instrument has been in operation since 2012. It is a successor to the initial HSRL instrument, described extensively by Hair et al. (2008); Burton et al. (2012), and is validated by Rogers et al. (2009). The instrument
uses the HSRL method in order to measure independently the extinction and backscatter coefficients of aerosol distributions at 355 nm (Burton et al., 2018) and 532 nm. Additionally, HSRL-2 utilises the standard backscatter technique for measuring the attenuated aerosol backscatter coefficient at 1064 nm, which is used to retrieve aerosol backscatter coefficient using Fernald (1984) with a minimum of error due to calibration against the 532 nm channel and the relative lack of attenuation at 1064 nm which makes the retrieval minimally sensitive to the unknown lidar ratio at 1064 nm (Müller et al., 2014). HSRL-2 measures
the depolarization ratio (Eq. 11) at all three lidar wavelengths. Thereby, the products returned from HSRL-2 are vertically resolved values of the backscatter coefficient and the depolarisation ratio at all three lidar wavelengths: 355, 532, and 1064 nm (Burton et al., 2015), and the extinction coefficient at the high-spectral-resolution channels 355 and 532 nm.

HSRL-2 was the first airborne lidar able to return the $3\alpha + 2\beta$ combination of measurements, which, as mentioned before, is essential for the retrieval of aerosol microphysical properties (Müller et al., 2014). For the ACEPOL flights, the HSRL
technique gives both extinction and backscatter. Extinction is measured with a vertical resolution of 150 m, over a 60 s period of flight. The elastic backscatter technique gives the backscatter coefficient, measured with a vertical resolution of 15 m, and a horizontal resolution of 1-2 km, depending on the speed of the aeroplane. The aerosol depolarisation ratio is measured with the same horizontal and vertical resolution as for the particulate backscatter, but here we work with values integrated over the total column.

## 270 4 Results

This section presents firstly the results of retrievals carried out with synthetic measurements, generated in the way as described in Sect. 3.1. Following this, the results from retrievals using measurements taken with HSRL-2 during the ACEPOL campaign are shown. In the synthetic measurements study, retrievals were carried out for synthetically-generated lidar measurements with no added noise, and with the inclusion of an instrumental noise model using the error setup described in Sect. 3.1.
Measurements with no added noise represent an ideal theoretical case, as any measurement will have a certain degree of noise contained within it. However, this set of results serves as a best case scenario, with a certain amount of degradation in retrieval quality evident as the noise is included. Plots showing the results of retrievals from an HSRL-2 setup with noise are presented in this subsection, along with tabulated results of the same instrumental setup without added noise. Additionally, in table form, we present retrieval metrics for the aforementioned super-lidar setup ($3\alpha + 3\beta + 3\delta^{\text{pol}}$), along with results for instrumental
setups corresponding to those of CPL and ATLID.





## 4.1 Synthetic data

Firstly, retrieval metrics from an HSRL-2 setup are presented, with scatter plots shown for the retrievals containing added measurement noise. Following this, results from the prior sensitivity study are presented, where the retrieval metrics are plotted as a function of prior for the various microphysical parameters. Finally, tables with retrieval metrics from the HSRL-2, super-

lidar, CPL, and ATLID setups, simplified as described above, with and without noise are presented. Figure 1 presents the results of retrieval versus truth for the microphysical parameters of each mode, for the case with added noise. Figure 2 shows the results for the AOD of each mode and the single scattering albedo. All metrics correspond to the 997 pixels meeting the retrieval criterion ($\chi^2 < 5$).

The fine-mode plots are shown first, with the effective radius having an MAE of 0.038 μm, a bias of 0.016 μm, and a corre-

lation coefficient of 0.698. The column number retrievals are plotted with logarithmic axes, with an MAE of $4.14 \times 10^{12}$ m$^{-2}$, a bias of -3.06$\times 10^{12}$ m$^{-2}$, and a correlation coefficient with a value of 0.793.

The retrievals of the spherical fraction have an MAE of 0.110, a bias of 0.043, and a correlation coefficient of 0.842. The scatter plot shows a dispersion above and below the line $y = x$, with an overestimation on average as shown by the bias, and a strong correlation between retrieval and truth. The real component of the refractive index has an MAE of 0.053 and a bias of

$2.31 \times 10^{-4}$. Finally in the fine-mode case, the logarithmic plot of the imaginary component of the refractive index shows an MAE of 0.010 and a bias of $9.12 \times 10^{-4}$. The correlation between the truth and retrieval for both real and imaginary refractive index components is rather poor, as exemplified by the $r$ values of 0.349 and 0.251, respectively.

Beneath the fine-mode plots, the corresponding results of the coarse-mode retrievals are shown. The retrievals of effective radius have an MAE of 0.459 μm, a bias of -0.107 μm, and a correlation coefficient of 0.686. The column number retrievals

have an MAE of $1.52 \times 10^{10}$ m$^{-2}$, a bias of -5.38$\times 10^9$ m$^{-2}$, and a correlation coefficient of 0.621. The spherical fraction retrievals have an MAE of 0.120, a bias of -0.087, and a correlation coefficient of 0.868, showing an overall underestimation but a strong correlation. The real component of the refractive index has an MAE of 0.085 and a bias of 0.071, demonstrating an overestimation on average. The last plot shows the imaginary component of the refractive index, on logarithmic axes, with an MAE of $1.71 \times 10^{-4}$ and a bias of $2.06 \times 10^{-4}$. The respective $r$ numbers are 0.522, and 0.573.

The final set of plots, Fig. 2, shows the AOD at 532 nm for each aerosol mode, along with the single scattering albedo at 532 nm. The MAE of the fine-mode AOD is 0.054, with a bias of 0.033, and a correlation coefficient of 0.962. The retrievals of the coarse-mode AOD have an MAE of 0.051, a bias of -0.033, and a correlation coefficient of 0.770. The fine-mode AOD is underestimated, as indicated by the negative bias, which is compensated by an overestimate in the coarse-mode AOD. The $r$ values indicate a strong correlation for both, especially in the fine-mode case. The single scattering albedo retrievals have

an MAE of 0.034 and a bias of $3.46 \times 10^{-3}$. The correlation coefficient of 0.375 indicates a poor relationship between the retrieved and the truth values.



**Figure 1.** Retrieved microphysical properties versus the truth values of both modes for the HSRL-2 setup with noise, for a total of 997 pixels from 1000 meeting the $\chi^2$ criterion. From top left to bottom right: fine-mode effective radius, fine-mode column number, fine-mode spherical fraction, fine-mode real refractive index at 550 nm, fine-mode imaginary refractive index at 550 nm, coarse-mode effective radius, coarse-mode column number, coarse-mode spherical fraction, coarse-mode real refractive index at 550 nm, and coarse-mode imaginary refractive index at 550 nm. Appended to the top left of each plot are the following numbers: the mean absolute error, the average bias, and the correlation coefficient. A positive bias indicates an average overestimation of the particular aerosol microphysical property, and a negative bias denotes that the parameter has been underestimated on average.





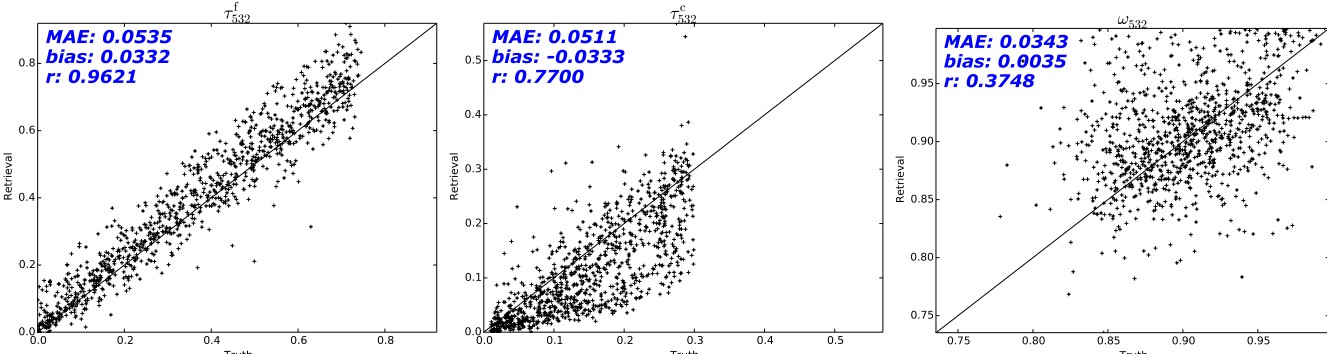

**Figure 2.** Retrieved optical properties versus the truth values at 532 nm for the HSRL-2 setup with noise. From left to right: AOD of the fine mode, AOD of the coarse mode, and single scattering albedo for the full aerosol distribution. Appended to the top left of each plot are the following numbers: the mean absolute error, the average bias, and the correlation coefficient. A positive bias indicates an average overestimation of the particular aerosol microphysical property, and a negative bias denotes that the parameter has been underestimated on average





Figure 3 shows the results from the prior sensitivity study, with the mean bias and MAE shown. Additionally, in the upper-left of each plot, the mean truth value of the parameter is shown. These plots are for the HSRL-2 setup with measurement noise. The top-left plot shows the results for $r_{\mathrm{eff}}^{\mathrm{f}}$, and the top right $r_{\mathrm{eff}}^{\mathrm{c}}$. Each plot shows that the bias approaches its lowest absolute

value near the mean truth, with the MAE also having its minimum point close to where the prior equals the truth. Both metrics grow in size as the prior deviates from the truth. The change in retrieval metrics is not as pronounced for prior values below the mean truth, indicating that choosing a prior that is too large for the effective radius leads to greater retrieval errors than a prior that is too small.

The middle two plots show the retrieval metrics for the spherical fraction, with the fine-mode results on the left-hand side

and the coarse-mode results on the right. For both aerosol modes, the bias approaches zero close to the mean truth value, with greater absolute values as the prior diverges to the parameter boundaries. The fine-mode retrieval metrics approach a minimum value close to the mean truth value, contrasted with the coarse-mode metrics which all tend to their lowest values as the prior is increased. This indicates that an overestimation of the coarse-mode spherical fraction will lead to a lower retrieval error than an underestimation.

The bottom-left plot shows the retrieval metrics for the real part of the fine-mode refractive index. This plot was achieved by varying the real part of the inorganic component of the refractive index, and keeping the black carbon component fixed, which kept the imaginary component constant. The bias approaches zero close to the mean truth value, as expected, and the MAE has its minimum near this point, with a greater value for an increased prior. The bottom-right plot illustrates the results of varying the prior for the imaginary component of the fine-mode refractive index. This experiment was conducted through

varying the imaginary part of the black carbon component, and keeping the inorganic component fixed, resulting in a constant real component. As for the real component, a bias near zero corresponding to the mean truth value is found, with the MAE close to its minimum at this value. The retrieval metrics rise as the prior is increased, which is consistent with that found for the real refractive index, and for the effective radius. Overall, the dependence of prior is relatively small for the real part of the refractive index, but stronger for the imaginary part. Analogous experiments were also carried out with the first guess, which

showed little variation compared to the experiments with a priori values.

Table 3 presents the bias, RMSE, MAE, and correlation coefficient for the HSRL-2 configuration with and without added measurement noise. Table 4 gives the retrieval metrics for the super-lidar configuration, using all nine optical parameters in the measurement vector, for cases with and without noise. Comparing the super-lidar and HSRL-2 noise-free cases, it is clear that the addition of an extra extinction channel in the measurement vector leads overall to a reduction in retrieval error, however

this is not apparent for the cases with noise. In the noise-free case, both the retrieved fine and coarse-mode effective radius have a lower bias and MAE for the super-lidar setup than in the HSRL-2 setup, along with a stronger correlation for the super-lidar retrievals. Additionally, the coarse-mode real refractive index has lower bias and MAE values in the super-lidar case, along with a stronger correlation than for the HSRL-2 retrievals. However, the difference between the super-lidar and HSRL-2 configuration is not so clear where measurement noise is included, as overall the results are quite similar in that case.

Table 5 shows the results from retrievals carried out using the simplified CPL setup, with and without added measurement noise. Table 6 gives the results from retrievals performed with the ATLID configuration. With the exception of the coarse-mode



**Figure 3.** Retrieval metrics as a function of prior, for the HSRL-2 setup with noise. Top plots: results for fine and coarse mode effective radius, with the prior in μm. Middle plots: results for fine and coarse-mode spherical fraction. Bottom plots: results for the fine-mode component of the real refractive index on the left, and results for the imaginary component of the fine-mode refractive index on the right. Each plot shows bias, and MAE.





**Table 3.** Results for retrievals with the $2\alpha + 3\beta + 3\delta^{\mathrm{pol}}$ HSRL-2 configuration with and without noise.

| | HSRL-2 without noise | | | | HSRL-2 with noise | | | |
|---|---|---|---|---|---|---|---|---|
| | Bias | RMSE | MAE | R | Bias | RMSE | MAE | R |
| $m_{\mathrm{r,\,550}}^{\mathrm{f}}$ | $-2.76 \times 10^{-3}$ | 0.049 | 0.037 | 0.492 | $2.31 \times 10^{-4}$ | 0.067 | 0.052 | 0.349 |
| $m_{\mathrm{i,\,550}}^{\mathrm{f}}$ | $-3.00 \times 10^{-4}$ | $9.44 \times 10^{-4}$ | $7.20 \times 10^{-3}$ | 0.463 | $7.35 \times 10^{-5}$ | 0.013 | 0.010 | 0.251 |
| $r_{\mathrm{eff}}^{\mathrm{f}}$ | 0.012 | 0.038 | 0.025 | 0.821 | 0.016 | 0.051 | 0.038 | 0.698 |
| $f_{\mathrm{sphere}}^{\mathrm{f}}$ | 0.037 | 0.117 | 0.071 | 0.922 | 0.043 | 0.163 | 0.109 | 0.842 |
| $\tau_{532}^{\mathrm{f}}$ | 0.028 | 0.055 | 0.039 | 0.978 | 0.033 | 0.071 | 0.054 | 0.962 |
| $m_{\mathrm{r,\,550}}^{\mathrm{c}}$ | 0.063 | 0.105 | 0.075 | 0.432 | 0.071 | 0.112 | 0.085 | 0.522 |
| $m_{\mathrm{i,\,550}}^{\mathrm{c}}$ | $1.57 \times 10^{-4}$ | $2.44 \times 10^{-4}$ | $1.90 \times 10^{-4}$ | 0.484 | $1.71 \times 10^{-4}$ | $2.57 \times 10^{-4}$ | $2.06 \times 10^{-4}$ | 0.573 |
| $r_{\mathrm{eff}}^{\mathrm{c}}$ | $-0.153$ | 0.364 | 0.254 | 0.909 | $-0.107$ | 0.630 | 0.459 | 0.686 |
| $f_{\mathrm{sphere}}^{\mathrm{c}}$ | $-0.071$ | 0.139 | 0.090 | 0.903 | $-0.089$ | 0.166 | 0.120 | 0.868 |
| $\tau_{532}^{\mathrm{c}}$ | $-0.029$ | 0.055 | 0.039 | 0.848 | $-0.033$ | 0.068 | 0.051 | 0.770 |
| $\omega_{532}$ | $2.30 \times 10^{-3}$ | 0.032 | 0.023 | 0.639 | $3.46 \times 10^{-3}$ | 0.046 | 0.034 | 0.375 |

refractive index components, the MAE and bias values of the noise-free setup of HSRL-2 are lower than those of CPL. This is also reflected in the greater values of correlation coefficient in all of these cases. When noise is added, the HSRL-2 results also show mostly lower values of MAE and bias compared to CPL, and a greater correlation coefficient. Contrasting the noise-free

retrievals for HSRL-2 and ATLID show HSRL-2 to have lower values of MAE and bias in most cases, along with a stronger correlation for all parameters, with this also the case when measurement noise is included.

Comparing the CPL and ATLID retrievals, the MAE values of ATLID have a lower retrieval error for the fine-mode parameters. The value of the correlation coefficient for ATLID shows a notably greater value, especially for the fine-mode spherical fraction. It should however be noted that CPL only measures the depolarisation ratio at 1064 nm, compared with ATLID's sin-

gular measurement at 355 nm. The retrievals of the coarse-mode for CPL mostly have lower values of MAE than for ATLID, probably owing to the inclusion of both depolarisation ratio and backscatter coefficient measurements at 1064 nm for CPL. The correlation coefficient is also larger for CPL than ATLID in the coarse-mode retrievals, with a strong correlation produced for the coarse-mode spherical fraction, and also the coarse-mode AOD.

### 4.2    ACEPOL data

In this subsection, lidar retrievals from real HSRL-2 measurements of particles in a smoke plume are presented. The lidar retrievals are performed on column-integrated lidar measurements, and hence are representative of an entire atmospheric column, in order to best facilitate comparison with polarimeter retrievals. Fu et al. (2020) have previously reported polarimetric retrievals of optical properties from measurements of this smoke plume, comparing retrievals from SPEX airborne and RSP. Here, we compare retrieved microphysical properties from HSRL-2 to the values reported by Fu et al. (2020). Retrievals were

carried out for fifteen individual pixels, with the error covariance matrix the same as used for the synthetic retrievals. Fu et al.



**Table 4.** Results for retrievals with the $3\alpha + 3\beta + 3\delta^{\text{pol}}$ super-lidar configuration with and without noise.

| | SL without noise | | | | SL with noise | | | |
|---|---|---|---|---|---|---|---|---|
| | Bias | RMSE | MAE | R | Bias | RMSE | MAE | R |
| $m_{\text{r, 550}}^{\text{f}}$ | $5.49 \times 10^{-3}$ | 0.049 | 0.036 | 0.568 | 0.015 | 0.086 | 0.066 | 0.286 |
| $m_{\text{i, 550}}^{\text{f}}$ | $-1.03 \times 10^{-3}$ | $8.83 \times 10^{-3}$ | $6.61 \times 10^{-3}$ | 0.611 | $9.12 \times 10^{-4}$ | 0.017 | 0.013 | 0.214 |
| $r_{\text{eff}}^{\text{f}}$ | $3.03 \times 10^{-3}$ | 0.025 | 0.016 | 0.918 | $8.90 \times 10^{-3}$ | 0.053 | 0.038 | 0.693 |
| $f_{\text{sphere}}^{\text{f}}$ | $9.12 \times 10^{-3}$ | 0.088 | 0.047 | 0.953 | 0.026 | 0.155 | 0.108 | 0.855 |
| $\tau_{532}^{\text{f}}$ | $5.99 \times 10^{-4}$ | 0.016 | 0.011 | 0.997 | $7.08 \times 10^{-3}$ | 0.054 | 0.040 | 0.970 |
| $m_{\text{r, 550}}^{\text{c}}$ | $1.54 \times 10^{-3}$ | 0.028 | 0.016 | 0.825 | 0.016 | 0.067 | 0.045 | 0.461 |
| $m_{\text{i, 550}}^{\text{c}}$ | $-5.14 \times 10^{-5}$ | $1.68 \times 10^{-4}$ | $1.26 \times 10^{-4}$ | 0.402 | $1.03 \times 10^{-4}$ | $2.86 \times 10^{-4}$ | $2.04 \times 10^{-4}$ | 0.264 |
| $r_{\text{eff}}^{\text{c}}$ | $-0.049$ | 0.258 | 0.156 | 0.948 | $-0.084$ | 0.641 | 0.470 | 0.682 |
| $f_{\text{sphere}}^{\text{c}}$ | $7.40 \times 10^{-4}$ | 0.035 | 0.018 | 0.992 | $-0.019$ | 0.108 | 0.070 | 0.928 |
| $\tau_{532}^{\text{c}}$ | $-2.69 \times 10^{-4}$ | 0.013 | $8.71 \times 10^{-3}$ | 0.988 | $-4.46 \times 10^{-3}$ | 0.037 | 0.026 | 0.909 |
| $\omega_{532}$ | $4.54 \times 10^{-3}$ | 0.027 | 0.020 | 0.752 | $4.51 \times 10^{-3}$ | 0.051 | 0.039 | 0.341 |

**Table 5.** Results for retrievals with the simplified CPL configuration with and without noise.

| | CPL without noise | | | | CPL with noise | | | |
|---|---|---|---|---|---|---|---|---|
| | Bias | RMSE | MAE | R | Bias | RMSE | MAE | R |
| $m_{\text{r, 550}}^{\text{f}}$ | 0.045 | 0.070 | 0.056 | 0.131 | 0.048 | 0.074 | 0.060 | 0.121 |
| $m_{\text{i, 550}}^{\text{f}}$ | $-6.62 \times 10^{-4}$ | $9.95 \times 10^{-3}$ | $8.50 \times 10^{-3}$ | 0.080 | $-7.26 \times 10^{-4}$ | 0.010 | $8.52 \times 10^{-3}$ | 0.082 |
| $r_{\text{eff}}^{\text{f}}$ | $7.37 \times 10^{-3}$ | 0.062 | 0.052 | $-0.065$ | 0.013 | 0.068 | 0.056 | -0.153 |
| $f_{\text{sphere}}^{\text{f}}$ | $2.74 \times 10^{-3}$ | 0.286 | 0.249 | 0.122 | $7.84 \times 10^{-3}$ | 0.288 | 0.251 | 0.100 |
| $\tau_{532}^{\text{f}}$ | $-0.052$ | 0.188 | 0.124 | 0.712 | $-0.054$ | 0.182 | 0.131 | 0.724 |
| $m_{\text{r, 550}}^{\text{c}}$ | 0.073 | 0.093 | 0.077 | 0.607 | 0.069 | 0.095 | 0.077 | 0.563 |
| $m_{\text{i, 550}}^{\text{c}}$ | $1.27 \times 10^{-4}$ | $1.51 \times 10^{-4}$ | $1.35 \times 10^{-4}$ | 0.821 | $1.24 \times 10^{-4}$ | $1.58 \times 10^{-4}$ | $1.38 \times 10^{-4}$ | 0.776 |
| $r_{\text{eff}}^{\text{c}}$ | $-0.498$ | 0.761 | 0.574 | 0.695 | $-0.441$ | 0.772 | 0.584 | 0.597 |
| $f_{\text{sphere}}^{\text{c}}$ | $-0.013$ | 0.145 | 0.111 | 0.873 | $-5.27 \times 10^{-3}$ | 0.158 | 0.120 | 0.841 |
| $\tau_{532}^{\text{c}}$ | $-0.029$ | 0.061 | 0.047 | 0.776 | $-0.022$ | 0.067 | 0.051 | 0.747 |
| $\omega_{532}$ | 0.011 | 0.039 | 0.030 | 0.268 | 0.012 | 0.041 | 0.032 | 0.204 |





**Table 6.** Results for retrievals with the Earthcare ATLID configuration with and without noise.. Note that the refractive indices would be the same for 355 nm, the wavelength at which ATLID is planned to operate.

| | ATLID without noise | | | | ATLID with noise | | | |
|---|---|---|---|---|---|---|---|---|
| | Bias | RMSE | MAE | R | Bias | RMSE | MAE | R |
| $m_{r,550}^f$ | $-0.011$ | $0.054$ | $0.043$ | $0.299$ | $-9.91 \times 10^{-3}$ | $0.056$ | $0.044$ | $0.282$ |
| $m_{i,550}^f$ | $2.58 \times 10^{-4}$ | $9.76 \times 10^{-3}$ | $8.27 \times 10^{-3}$ | $0.153$ | $2.48 \times 10^{-4}$ | $9.84 \times 10^{-3}$ | $8.36 \times 10^{-3}$ | $0.142$ |
| $r_{eff}^f$ | $0.010$ | $0.058$ | $0.049$ | $0.318$ | $8.78 \times 10^{-3}$ | $0.059$ | $0.050$ | $0.297$ |
| $f_{sphere}^f$ | $0.019$ | $0.224$ | $0.177$ | $0.619$ | $0.025$ | $0.226$ | $0.179$ | $0.621$ |
| $\tau_{532}^f$ | $0.070$ | $0.131$ | $0.094$ | $0.918$ | $0.067$ | $0.138$ | $0.099$ | $0.910$ |
| $m_{r,550}^c$ | $0.096$ | $0.114$ | $0.101$ | $0.388$ | $0.090$ | $0.114$ | $0.101$ | $0.407$ |
| $m_{i,550}^c$ | $1.75 \times 10^{-4}$ | $2.12 \times 10^{-4}$ | $1.86 \times 10^{-4}$ | $0.468$ | $1.66 \times 10^{-4}$ | $2.11 \times 10^{-4}$ | $1.84 \times 10^{-4}$ | $0.484$ |
| $r_{eff}^c$ | $-0.380$ | $0.827$ | $0.666$ | $0.372$ | $-0.397$ | $0.838$ | $0.673$ | $0.353$ |
| $f_{sphere}^c$ | $3.51 \times 10^{-3}$ | $0.257$ | $0.218$ | $0.392$ | $-4.35 \times 10^{-3}$ | $0.256$ | $0.216$ | $0.423$ |
| $\tau_{532}^c$ | $-0.063$ | $0.090$ | $0.072$ | $0.653$ | $-0.058$ | $0.095$ | $0.076$ | $0.564$ |
| $\omega_{532}$ | $-3.43 \times 10^{-4}$ | $0.035$ | $0.028$ | $0.409$ | $3.83 \times 10^{-4}$ | $0.036$ | $0.029$ | $0.400$ |

(2020) show the flight path of the aircraft during this part of the measurement campaign. We found that the retrieval was quite sensitive to the choice of the first guess for the fine-mode effective radius, so we performed retrievals for several choices of first guess, and for each pixel chose the first guess that resulted in the smallest $\chi^2$ between forward model and measurement. The first guess of the other microphysical parameters was set to the result of the SPEX airborne retrieval. The prior values of the
microphysical parameters were fixed to the middle of the range. Out of the fifteen pixels, ten were found to meet the criterion for a successful retrieval, $\chi^2 \leq 5$.

The median and standard deviation of microphysical parameters and optical properties are presented in Table 7, along with the values from Fu et al. (2020) of SPEX and RSP retrievals shown additionally for comparison. As in Fu et al. (2020), results are only shown for pixels with $\tau_{532} > 0.2$, to have reasonable sensitivity to the retrieval of microphysical properties. The
coarse-mode contribution to the measurements is negligible, thus only the fine-mode microphysical properties are presented. The HSRL-2 retrieval of the real component of the refractive index is in good agreement with both polarimeter retrievals, although the imaginary component is not as close to that of the polarimeter results. This is also expected from the synthetic retrievals, that show poor capability of lidar measurements to constrain absorption.

The values for effective radius between lidar and polarimeter retrievals are in close agreement, with the lidar retrievals
showing a larger standard deviation, perhaps illustrative of the greater difficulty in accurately retrieving the effective radius for the lidar measurements. This is also expected from the synthetic experiment. For example, the MAE in Table 3 for $r_{eff}^f$ is much larger that what is expected from polarimeters (Hasekamp et al., 2019a). The retrieved fine-mode AOD from the HSRL-2 measurements at 532 nm is close to that of the polarimeter retrievals, with a slightly lower standard deviation than for the polarimeter results. The retrieved value of the coarse-mode AOD lies within the range of the SPEX and RSP results. The





**Table 7.** Results of the HSRL-2 smoke-plume retrievals, along with the retrieved values from SPEX and RSP from Fu et al. (2020) for comparison.

|  | HSRL-2 | | SPEX | | RSP | |
| --- | --- | --- | --- | --- | --- | --- |
|  | Median | SD | Median | SD | Median | SD |
| Fine-mode real component of the refractive index ($m_{\mathrm{r,\,532}}^{\mathrm{f}}$) | 1.561 | 0.030 | 1.579 | 0.019 | 1.556 | 0.059 |
| Fine-mode imaginary component of the refractive index ($m_{\mathrm{i,\,532}}^{\mathrm{f}}$) | 0.010 | 0.011 | 0.038 | 0.011 | 0.036 | 0.013 |
| Fine-mode effective radius ($r_{\mathrm{eff}}^{\mathrm{f}}$, in μm) | 0.107 | 0.053 | 0.116 | 0.004 | 0.119 | 0.007 |
| Fine-mode AOD ($\tau_{532}^{\mathrm{f}}$) | 0.505 | 0.202 | 0.554 | 0.238 | 0.509 | 0.231 |
| Coarse-mode ($\tau_{532}^{c}$) | 0.021 | 0.007 | 0.016 | 0.011 | 0.040 | 0.029 |
| SSA ($\omega_{532}$) | 0.940 | 0.086 | 0.815 | 0.044 | 0.829 | 0.044 |
| Fraction of spherical particles ($f_{\mathrm{sphere}}$) | 0.719 | 0.232 | 0.989 | 0.149 | 0.846 | 0.133 |

single-scattering albedo, as for the imaginary component of the refractive index, does not reflect the same amount of absorption inferred from the polarimeter retrievals. This is expected, because it is known that lidar measurements do not give the same insight into absorption properties as multi-angle spectropolarimetric measurements. The HSRL-2 retrieval of the fraction of spherical particles is lower than both polarimeter retrievals, though it is in line with the difference between SPEX and RSP, and what is to be expected from biomass burning, see for example Nicolae et al. (2013).

## 5 Summary and conclusion

In this study, an investigation into retrieving aerosol microphysical properties from multi-spectral lidar measurements has been presented. Aerosol retrievals with a purpose-built algorithm were performed with a bimodal setup, in order to aid direct comparison with polarimetric retrievals. Firstly, experiments with synthetic measurements were carried out, in order to fully gauge the capability of retrieving microphysical properties from a bimodal aerosol distribution. The synthetic measurements

study shows that with a $2\alpha + 3\beta + 3\delta^{\mathrm{pol}}$ lidar setup corresponding to that of the HSRL-2 instrument, reasonable retrievals of microphysical properties of a bimodal aerosol ensemble can be performed. Specifically, this setup has the capability to retrieve information on fine and coarse-mode effective radius, column number, and spherical fraction. In many cases, the fine-mode real component of the refractive index is also retrieved with a close match to the truth values, albeit with a relatively large spread. Adding noise to the measurement vector results in a degradation of retrieval quality, as evident from the retrieval metrics shown,

though of course a setup inclusive of noise more closely replicates that of actual lidar measurements from the atmosphere or from space. Information can also be gleaned from retrievals of the fine and coarse-mode AOD.

As well as simulating lidar measurements for HSRL-2, also investigated was the increase of the number of input parameters, and how this could improve retrieved quantities. This was carried out by adding the extinction at 1064 nm to the measurement vector of the synthetic measurements, which is not measured with HSRL-2. For the case with no added measurement error,

raising the number of measurements from eight to nine values lead to a reduction of the MAE and absolute value of bias in



most cases, clearly evident in the retrievals of effective radius and spherical fraction for both fine and coarse modes. In the cases with added noise, it appears that the measurement noise diminishes any further constraint on the microphysical parameters that the extra value of extinction could potentially give.

Also tested was a configuration akin to that of a simplified CPL setup, with the experiments showing that only using a
four component measurement vector results in significantly worse aerosol retrievals, as expected. Additionally, retrievals were performed using a measurement setup analogous to that of the monospectral ATLID instrument. A comparison to the results from HSRL-2 show the latter is better suited to retrieving information on the microphysical parameters. This is evident from the lower MAE in most cases for HSRL-2, and the stronger correlation for all of the retrieved parameters.

In the second part of this study the retrieval algorithm was applied to lidar measurements from HSRL-2 taken during the
ACEPOL campaign. These measurements were taken over a prescribed forest fire in Arizona, with large AOD values (including AOD > 1). The ACEPOL HSRL-2 retrievals yielded an effective radius and a real component of refractive index close to those of the polarimeter retrievals, in line with that expected from biomass burning. The difference in the retrieved spherical fraction is close to the difference between the SPEX and RSP retrievals from Fu et al. (2020). The imaginary component of the refractive index and consequently the single-scattering albedo proved to be underestimated in comparison to the polarimeter retrievals,
though lidar measurements are generally not expected to be able to fully characterise the absorption properties of an aerosol ensemble, since retrieved absorption properties are strongly dependent on the prior.

Overall, this paper has presented an investigation of the quality of aerosol microphysical retrievals from air or space-based lidar measurements with a bimodal approach. The capability of an instrumental setup similar to that of HSRL-2 was investigated, along with a super-lidar setup containing one additional lidar measurement. A setup with nine individual lidar measurements,
namely: AOD, depolarisation ratio, and backscatter coefficient at three wavelengths, appears to be the most optimal configuration in the framework of this study, though when taking into account the concomitant error associated with a measurement, the super-lidar setup does not yield more information than that of the HSRL-2 setup. Finally, applying the retrieval algorithm to data measured by HSRL-2 during the ACEPOL campaign showed the capabilities of the algorithm in conducting retrievals of data with a relatively high AOD from biomass burning. The lidar retrievals of the smoke-plume data yielded results close to
those of the polarimeter retrievals for some of the microphysical parameters. The effective radius retrieval was a close match to the SPEX result, with a good match in the real component of the fine-mode refractive index. This work has considered a single atmospheric layer in the experiments that are presented, which can be taken as representative of either the entire atmospheric column below an aircraft or satellite, or a single atmospheric layer of fixed vertical dimension. A future study will investigate the combined effectiveness of aerosol retrievals utilising both lidar and polarimeter measurements.

*Author contributions.* WGKM and OPH designed the experiment and carried out the research. GF provided the column-integrated HSRL-2 data and contributed to definition of the retrieval algorithm. SPB provided invaluable feedback that substantially improved the paper, and was involved in operations and calibration processing of HSRL-2. OPH contributed to the campaign definition and flight planning. WGKM and OPH wrote the paper.



*Competing interests.* The authors declare that they have no conflict of interest.

*Acknowledgements.* This work is funded by the NWO/NSO project ACEPOL (Aerosol Characterization from Polarimeter and Lidar) under project number ALW-GO/16-09. We express our gratitude to everyone involved in the ACEPOL campaign. We acknowledge the former Aerosol, Cloud, Ecosystem (ACE) program at NASA's Earth Science Division as a sponsor for ACEPOL flights. We thank the field teams making measurements on the ground as some of those were critical to the measurement accuracy (i.e., vicarious calibration). We also acknowledge the support received by the ground and air crew at NASA AFRC in Palmdale.



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
