# Peer review of "Retrieval of aerosol microphysical properties from atmospheric lidar sounding: an investigation using synthetic measurements and data from the ACEPOL campaign"

_Atmospheric Measurement Techniques, 2021_

## Author Comment (AC1)

**Response to comments**

Authors suggest an approach to inversion of multiwavelength lidar measurements to the particle microphysical parameters based on iteration scheme with prior assumption about particle properties. In this manuscript authors make an important step considering parameters both the fine and the coarse mode. Simulation performed with synthetic data provides estimation of retrieval uncertainties for different lidar configurations. The manuscript is well and clearly written and matches AMT scientific criteria.

I have just several short comments.

Authors consider 12 independent parameters of aerosol, when even for "super-lidar" only 9 observations are available. The problem is underdetermined and unique solution does not exist. I think this principal question should be discussed in the beginning of the manuscript.

This becomes especially critical when configuration corresponding CPL or ATLID lidar are considered.

*Thanks, we do indeed need to further emphasise the ill-posed nature of the problem, and the underdeterminedness of the system, and the motivation of our experiments, which is to investigate where the information in the lidar measurements goes in an iterative retrieval. To that end, around line 110 in the introduction we have changed the description of the method as such: "Keeping in mind the results of uncertainty/information content analysis, we apply an iterative retrieval scheme, taking the lidar measurements and investigating where the information in the measurements goes in a retrieval of microphysical parameters. The problem is clearly ill-posed, and the system underdetermined, as the number of microphysical parameters we attempt to retrieve can be around twice the number of measurements, depending on the configuration"*

*Additionally, we have included the following in the summary and conclusion, in the first paragraph: "For the HSRL-2 configuration, the three measurements of depolarisation ratio yield information on the spherical fraction of the aerosol distribution. This leaves the two extinction measurements and the three backscatter measurements to provide information on the remaining microphysical properties. The problem is clearly ill-posed, and the system is underdetermined, with the number of unknowns exceeding the number of measurements by almost a factor of two. Thus, it is clear that a prior is needed to provide a constraint on the microphysical properties of the aerosol distribution."*

Choose of model refractive indices for different types of aerosol is somehow convenient, but question is how sensitive are results to the choice of model. For example, in Table 1 the imaginary part of dust at 355 nm is 1.66*E-2. The same time, in recent study the Im for dust at 370 nm is below 0.005 (Di Biagio, Atmos. Chem. Phys., 19, 15503–15531, 2019) for dust of different origin. Will it influence the inversion?

*The influence of different refractive indices is outside of the scope of this study, as the aim was to carry out aerosol retrievals in a manner aligned as closely as possible to those carried out previously in polarimeter retrievals (e.g. in Fu et al. 2020)). As such, we used the same refractive index values as in Fu et al., as this is also crucial for the next part of the study involving a combination of lidar and polarimeter in a retrieval. There is an option to separately fit the imaginary and real refractive index spectra, which leaves more room to adjust the imaginary part whilst keeping the real component fixed.*

In Table 7 the parameters of the coarse mode obtained from the lidar measurements are not provided. Any reason?

*The coarse mode contribution is negligible compared to the smoke-dominated fine mode (fine mode AOD 24 times higher than coarse mode AOD). The AOD of the coarse mode retrieved from the ACEPOL HSRL-2 measurements is 0.021. For such low mode AOD it is virtually impossible to retrieve meaningful information on the microphysical properties of that mode.*

Fig.1. Axis title fonts should be increased.

*Thanks, we have enlarged the fonts and improved the presentation of all figures.*

**References**

Fu, G., Hasekamp, O., Rietjens, J., Smit, M., Di Noia, A., Cairns, B., Wasilewski, A., Diner, D., Seidel, F., Xu, F., Knobelspiesse, K., Gao, M., da Silva, A., Burton, S., Hostetler, C., Hair, J., and Ferrare, R.: Aerosol retrievals from different polarimeters during the ACEPOL campaign using a common retrieval algorithm, Atmospheric Measurement Techniques, 13, 553–573, https://doi.org/10.5194/amt-13- 553-2020, 2020.

---

## Author Comment (AC2)

**Response to comments**

Authors presented a study of aerosol microphysical retrievals from HSRL-2 instrument using both simulated data and real lidar measurements. The used iterative algorithm is based on Phillips-Tikhonov regularization. It is very unfortunate for the algorithm that quality of its results depends on the choice of prior and first guess. Retrieval errors increase when the prior deviates too much from the truth value that is a disadvantage for the real lidar data processing. At the moment, there is no way to have a good first guess for parameters like complex refractive index (especially, its imaginary part) or column number in the mass production mode.

*We would like to thank the reviewer for his/her important comments. We agree that it is unfortunate that the results depend on the choice of prior. However, this is not a drawback of the retrieval algorithm but rather the intrinsically limited information content of the lidar measurements.*

Overall, this paper is well written and can be published after minor corrections:

- I would suggest renaming the "alpha" variable (mode component coefficient) in Eq. 2. Throughout the paper, "alpha" means aerosol extinction and only in Eq. 2 there is a confusing turn.

*I have renamed this variable "$C_m$".*

- There is no need to use blue color in figures at all. The blue curves in Fig. 3 can be dashed to be more friendly to people who have access only to black-and-white printer.

*The blue coloured curves have been changed to dashed lines.*

- Line 105 says: "ii) Instead of performing a linear uncertainty/information content analysis, we apply an iterative retrieval scheme, that can be used to perform actual retrievals." The use of word "Instead" makes me think that the "iterative retrieval scheme" is somehow better compared to the "information content analysis" and have to replace the information content analysis. These two things are doing completely different job and can't be done instead of each other. I would say something like "Keeping in mind the results of uncertainty/information content analysis, we apply an iterative retrieval scheme..." to avoid the negative connotations of the word "instead".

*Thanks for pointing this out. Indeed we do not suggest our retrieval method should be compared to information content analysis. This has now been changed to "Keeping in mind the results of uncertainty/information content analysis, we apply an iterative retrieval scheme…" as you suggest.*

Now let's talk about the results of information content analysis.

Authors mentioned that "$3\beta + 2\alpha$ combination of measurements provides about 3-4 independent pieces of information" (line 100). Also, Authors are saying that "In addition to the underdeterminedness of the inversion problem, limitations to the forward model can further inhibit the retrieval of the microphysical properties" (line 165). At the same time

"With this parameterization, the total number of parameters free to vary in the retrieval is six for each mode, thereby twelve in total for the bimodal setup we use" (line 150). Let's forget about dust and consider only spherical particles. Depolarization is zero and useless in this case. So, $3\beta + 2\alpha$ lidar measures 5 numbers that gives Authors only 3-4 independent pieces of information. At the bottom line, Authors would like to use 3-4 pieces of information to retrieve 12 independent parameters and it sounds like a miracle. It is clear that the inverse solution is not unique, and Authors experience massive issues in Their retrievals, but Authors don't want to talk about it. I would like Authors to directly acknowledge this issue in Their paper. It can be done be adding several solid sentences clearly discussing the non-uniqueness of inverse solution due to the simple 3-4/12 math and the ways that Authors offer to increase the information content. Iterative algorithm by itself is not increasing the information content at all.

*This is an important point, yes, one that we do not emphasise clearly enough. Excluding the depolarisation ratio as we get only the spherical fraction from this, we then have the six $3\beta + 3\alpha$ lidar measurements for the "super-lidar", from which we attempt to retrieve the 10 microphysical parameters. Our results suggest that the majority of information from these measurements goes to the effective radius, the column number, and the real component of refractive index of the dominant mode. The results of the retrieval for the imaginary component of the refractive index and effective variance demonstrate that the measurements do not contain much information to constrain these, most so in the case of effective variance. The effective variance plots and metrics were omitted for brevity, as in none of our retrieval setups it could be constrained (basically, it fluctuates around the prior). The results for effective variance from the HSRL-2 setup with noise are shown in Fig. 1 of this reponse to demonstrate how poorly it is retrieved.*

[Figure]

Figure 1: Effective variance for the fine mode (left) and the coarse mode (right) for the HSRL-2 setup with measurement noise.

*Around line 110, in the introduction, we have changed the text thusly: "Keeping in mind the results of uncertainty/information content analysis, we apply an iterative retrieval scheme, taking the lidar measurements and investigating where the information in the measurements goes in a retrieval of microphysical parameters. The problem is clearly ill-posed, and the system underdetermined, as the number of microphysical parameters we attempt to retrieve can be around twice the number of measurements, depending on the configuration."*

*At the end of the first paragraph of the results section we have added: "We do not show the effective variance plots, nor do we include the results in the tables later in this section. This is because it is the least-well retrieved parameter out of the six microphysical parameters for each aerosol mode, with a strong dependence on the choice of prior, and no correlation shown between the truth value and the retrieved value in any of the measurement configurations trialled. It is clear that the lidar measurements we use do not contain sufficient information to provide constraints on the effective variance."*

*Additionally, we have included the following in the summary and conclusion, in the first paragraph: "For the HSRL-2 configuration, the three measurements of depolarisation ratio yield information on the spherical fraction of the aerosol distribution. This leaves the two extinction measurements and the three backscatter measurements to provide information on the remaining microphysical properties. The problem is clearly ill-posed, and the system is underdetermined, with the number of unknowns exceeding the number of measurements by*

*almost a factor of two. Thus, it is clear that prior information is needed to provide a constraint on the microphysical properties of the aerosol distribution."*

*Finally, added to the second paragraph in the summary and conclusion section: "The majority of information provided by the lidar measurements goes to these microphysical properties, with a preference for the dominant mode, and it appears little-to-no information is provided for the imaginary component of refractive index or the effective variance."*

I also would suggest: Line 5 "high spectral resolution lidar (HSRL)" instead of "High Spectral Resolution Lidar (HSRL)"

*We have changed this as you suggest.*

Line 15 "higher aerosol optical depth (AOD)" instead of "higher AOD"
*We have changed this as you suggest.*

Line 20 - MAEs are shown for all the input parameters except effective variance. Effective variance also deserves to be shown here. Please show the effective variance in all the tables with results (if available).

*For all of the configurations, the retrieval of effective variance generates results akin to those of Fig. 1. We believe that it is more prudent to show the errors for microphysical parameters for which the information from the measurements goes to in the inversion.*

- Please include also relative errors for effective radius and effective variance [0.038/??% (0.025/??%)].

*Perhaps it is best instead to include the average truth value of the effective radius for each mode: 0.195 micron for the fine mode, and 1.970 micron for the coarse mode. Thus I have changed this part of the abstract to: "The synthetic HSRL-2 retrievals resulted for the fine mode in a mean absolute error (MAE) of 0.038 (0.025)* $\mu$m *for the effective radius (with a mean truth value of 0.195* $\mu$m*), 0.052 (0.037) for the real refractive index, 0.010 (7.20$\times$10$^{-3}$) for the imaginary part of the refractive index, 0.109 (0.071) for the spherical fraction, and 0.054 (0.039) for the AOD at 532 nm, where the retrievals inside brackets indicate the MAE for noise-free retrievals. For the coarse mode, we find the MAE is 0.459 (0.254)* $\mu$m *for the effective radius (with a mean truth value of 1.970* $\mu$m*), 0.085 (0.075) for the real refractive index, 2.06$\times$10$^{-4}$ (1.90$\times$10$^{-4}$) for the imaginary component, 0.120 (0.090) for the spherical*

*fraction, and 0.051 (0.039) for the AOD. "*

Line 30 "Spectropolarimeter for Planetary Exploration (SPEX) and Research Scanning Polarimeter (RSP)" instead of "SPEX and RSP".

*We have changed this as you suggest.*

Line 50 "AOD and SSA" instead of "Aerosol Optical Depth (AOD) and Single Scattering Albedo (SSA)".

*We have changed this as you suggest.*

Line 60 - Acronym "POLDER" needs to be defined.

*This now reads: "...performed with the POLarization and Directionality of the Earth's Reflectances (POLDER) instrument."*

- "Measurements using HSRL techniques" instead of "Measurements using high spectral resolution lidar (HSRL) techniques".

*Changed as you suggest.*

Line 75 Acronym "ATLID" needs to be defined.

*ATLID is now defined.*

Line 90 "the $3\beta + 2\alpha$ setup" instead of "the $3\alpha + 2\beta$ setup".

*Thanks, we have now corrected this.*

Line 100 "used in MAP retrievals" instead of "used in Multi-Angle Polarimeter (MAP) retrievals".

*We have changed this as you suggest.*

Line 110 "during the ACEPOL campaign" instead of "during the Aerosol Characterization from Polarimeter and Lidar (ACEPOL) campaign".

*We have changed this as you suggest.*

Line 115 "onboard (CPL (McGill et al., 2002))" instead of "onboard (Cloud Physics Lidar (CPL) (McGill et al., 2002))".

*We have changed this as you suggest.*

Line 120 "(i.e. AOD)" instead of "(i.e. the aerosol optical depth (AOD))".

*We have changed this as you suggest.*

Line 220 "from HSRL measurements" instead of "from high spectral resolution lidar measurements".

*We have changed this as you suggest.*

Line 225
"and the RSP" instead of "and the Research Scanning Polarimeter (RSP)".

*We have changed this as you suggest.*

Figure 1 Please add the plots for effective variance and remove the redundant "Retrieval" in the right two columns to increase the font size. The font size in plots is normally 2 pt smaller compared to the main text.

*As mentioned in the response to your earlier point regarding the inclusion of the effective variance plots, we have instead made clear in the text that we choose not to include this and explain that the retrieval extracts very little information on effective variance from the measurements. We hope the extra sentences we have included in response to your earlier comment provide a sufficient justification for omitting the effective variance.*

Figure 3 "Metric" is redundant in the right column of plots and can be removed to increase the font size.

*We have changed this as you suggest.*

Tables 3-6 Effective variance is definitely missing here.

*We hope that the extra points included in the paper are sufficient to justify the omission of effective variance.*

---

## Author Comment (AC3)

**Response to comments**

The authors describe an iterative algorithm for the retrieval of aerosol microphysical properties from atmospheric lidar sounding based on damped Gauss-Newton method including Tikhonov-Phillips regularization. They assume a bimodal-log-normal distribution of a mixture of spheroids and spherical particles. First a simulation study is done. Finally a measurement case is evaluated and compared with the SPEX and RSP retrievals.

The results are interesting, although a few limitations, questions and drawbacks are open. List of remarks in random order:

The under-determinedness of the retrieval yields in non-uniqueness. The authors observed the following property:

"A study of the sensitivity of retrievals to the choice of prior and first guess showed that, on average, the retrieval errors increase when the prior deviates too much from the truth value." This is a well-known phenomenon of iterative methods in case of non-uniqueness. Depending on the initial value the algorithm converges to another "solution". This is the main drawback of the presented method. Question: How many iteration steps were made until the algorithm stops?

It is important to distinguish between the effect of prior and the effect of 1st guess. We do not see a large dependence of the retrieval outcome on the first guess, i.e. the algorithm does not converge to another solution depending on the initial values. This means the iteration approach works well and poses no limitation for the retrieval. On the other hand, there is a clear dependence on prior. This is not related to the retrieval scheme but to the ill-posed nature of the problem. It is clear that we need to further emphasise the underdetermined nature of the system, and the ill-posed nature of the inversion. We have changed the text around line 110 as such: "Keeping in mind the results of uncertainty/information content analysis, we apply an iterative retrieval scheme, taking the lidar measurements and investigating where the information in the measurements goes in a retrieval of microphysical parameters. The problem is clearly ill-posed, and the system underdetermined, as the number of microphysical parameters we attempt to retrieve can be around twice the number of measurements, depending on the configuration."

Additionally, we have included the following in the summary and conclusion, in the first paragraph: "For the HSRL-2 configuration, the three measurements of depolarisation ratio yield information on the spherical fraction of the aerosol distribution. This leaves the two extinction measurements and the three backscatter measurements to provide information on the remaining microphysical properties. The problem is clearly ill-posed, and the system is underdetermined, with the number of unknowns exceeding the number of measurements by almost a factor of two. Thus, it is clear that prior information is needed to provide a constraint on the microphysical properties of the aerosol distribution."

Finally, added to the second paragraph in the summary and conclusion section: "The majority of information provided by the lidar measurements goes to these microphysical properties, with a preference for the dominant mode, and it appears little-to-no information is provided for the imaginary component of refractive index or the effective variance."

We used a maximum of 30 iteration steps, with the state vector and Jacobian updated in each step.

Please, explain all abbreviations.

We have improved our presentation of abbreviated quantities.

"The coarse-mode contribution to the measurements is negligible, thus only the fine- mode microphysical properties are presented". Why? Please, include the values.

The coarse mode contribution is negligible compared to the smoke-dominated fine mode (fine mode AOD 24 times higher than coarse mode AOD). The AOD of the coarse mode retrieved from the ACEPOL HSRL-2 measurements is 0.021. For such low mode AOD it is virtually impossible to retrieve meaningful information on the microphysical properties of that mode.

Line 145: I am wondering about the variable n which was not introduced before.

Thanks for pointing this out, n is now defined as such: "...state vector, and  $n_{C_m}^{f;c}$  is the number of coefficients for the fine or coarse mode." Additionally, for clarity, we now use  $C_m$  to represent the refractive index coefficients instead of  $\alpha$ , as  $\alpha$  features as the symbol for extinction coefficient throughout our work.

(8) and (9): I am wondering that MAE is the same as bias?

For calculating the MAE, we take the absolute value of the difference between truth and retrieval, wheres the bias does not take into consideration the absolute difference, which allows us to determine whether a parameter is underestimated or overestimated. Equations 8 and 9 in the paper show the MAE and bias, respectively, with the MAE given by:

$$MAE = \frac{1}{n_{\text{pass}}} \sum_{i=1}^{n_{\text{pass}}} |x_i^{\text{retr}}[j] - x_i^{\text{truth}}[j]|, \qquad (1)$$

and the bias:

bias =
$$\frac{1}{n_{\text{pass}}} \sum_{i=1}^{n_{\text{pass}}} (x_i^{\text{retr}}[j] - x_i^{\text{truth}}[j]), \qquad (2)$$

"The correlation between the truth and retrieval for both real and imaginary refractive index components is rather poor, as exemplified by the r values of 0.349 and 0.251, respectively." This was observed even in retrieval techniques for spherical particles, see Mueller et al, AMT 9 (2016) 5007-5035. The authors should compare their simulation studies with those techniques.

At line 305 we have added: "Previous work considering spherical particles has also shown that it is challenging to retrieve the complex refractive index from lidar measurements (Müller et al., 2016)."

The pt-font in all Fig. is too small.

We have now rectified this.

The main results of Tables 3-6 should be summarized, additionally, in a Figure for the conveniences of the readers. The presentation is boring.

The plots in Fig. 1 of this response have been added to visualise the differences in bias and MAE between the different instrumental setups.

Table 7 caption: The authors should provide more information about:" ...what is to be expected from biomass burning, see for example Nicolae et al. (2013)"

We have added the following to the paper at line 400: As discussed in Fu et al. (2020), the SPEX and RSP values are commensurate with those expected for a smoke plume, specifically that smoke is comprised mainly of fine-mode particles (e.g. Russell et al. (2014)), which is shown by the difference in the retrieved fine-mode and coarse-mode AOD values. The

Figure 1: LHS: bias for each of the four instruments, for the various parameters. RHS: corresponding plot of MAE.

clear dominance of the fine mode is also evident from the values of AOD retrieved from the HSRL-2 measurements. The real component of the refractive index retrieved from HSRL-2 is consistent with those reported by Levin et al. (2010), from the Fire Laboratory at Missoula Experiment (FLAME), where the real refractive index value corresponding to biomass burning was found to be mostly in the range 1.55 to 1.60. Additionally, the SSA value for biomass burning was found by Nicolae et al. (2013) to have a value of 0.79 for smoke at 532 nm, with an age of 6 hours, and 0.93 for smoke aged 18 hours. The values retrieved by SPEX and RSP can be considered realistic for smoke, however as mentioned previously we do not expect a better retrieval of absorption properties from HSRL-2 than from MAPs.

It was interesting to learn that using the presented algorithm one gets: "However, the difference between the super-lidar and HSRL-2 configuration is not so clear where measurement noise is included, as overall the results are quite similar in that case." This means in case of measurement errors more input information does not result in a more accurate retrieval. May be this regularization method is not the best one for this retrieval or the regularization parameter was not selected appropriate? Please, regard this point.

The most likely explanation for the fact that the  $3\alpha + 3\beta$  yields similar results to  $3\alpha + 2\beta$  is that the information from the extra measurement in the latter case has large overlap with the information already in the other 5 measurements, i.e. it adds only little new information. This small amount of new information seems to 'drown' within the noise for the case where noise is added. Given that we allow for a flexible choice of regularisation parameter, it is unlikely that an inappropriate choice is the cause of the similar performance between  $3\alpha + 3\beta$  and  $3\alpha + 2\beta$ .

**References**

Fu, G., Hasekamp, O., Rietjens, J., Smit, M., Di Noia, A., Cairns, B., Wasilewski, A., Diner, D., Seidel, F., Xu, F., Knobelspiesse, K., Gao, M., da Silva, A., Burton, S., Hostetler, C., Hair, J., and Ferrare, R.: Aerosol retrievals from different polarimeters during the ACEPOL campaign using a common retrieval algorithm, Atmospheric Measurement Techniques, 13, 553–573, 2020.

Müller, D., Böckmann, C., Kolgotin, A., Schneidenbach, L., Chemyakin, E., Rosemann, J., Znak, P., and Romanov, A.: Microphysical par- ticle properties derived from inversion algorithms developed in the framework of EARLINET, Atmospheric Measurement Techniques, 9, 5007–5035, 2016.

Russell, P. B., Kacenelenbogen, M., Livingston, J. M., Hasekamp, O. P., Burton, S. P., Schuster, G. L., Johnson, M. S., Kno- belspiesse, K. D., Redemann, J., Ramachandran, S., and Holben, B.: A multiparameter aerosol classification method and its application to retrievals from spaceborne polarimetry, Journal of Geophysical Research: Atmospheres, 119, 9838–9863, 2014.

Levin, E. J. T., McMeeking, G. R., Carrico, C. M., Mack, L. E., Kreidenweis, S. M., Wold, C. E., Moosmüller, H., Arnott, W. P., Hao, W. M., Collett Jr., J. L., and Malm, W. C.: Biomass burning smoke aerosol properties measured during Fire Laboratory at Missoula Experiments (FLAME), Journal of Geophysical Research: Atmospheres, 115, 2010.

Nicolae, D., Nemuc, A., Müller, D., Talianu, C., Vasilescu, J., Belegante, L., and Kolgotin, A.: Characterization of fresh and aged biomass burning events using multiwavelength Raman lidar and mass spectrometry, Journal of Geophysical Research (Atmospheres), 118, 2956–2965, 2013.